# Optimum hospice at home services for end-of-life care: protocol of a mixed-methods study employing realist evaluation

Claire Butler,[1] Charlotte Brigden,[2] Heather Gage,[3] Peter Williams,[4] Laura Holdsworth,[5] Kay Greene,[6] Bee Wee,[7] Stephen Barclay,[8] Patricia Wilson[1]

[1]Centre for Health Services Studies, University of Kent, Canterbury, UK
[2]Pilgrims Hospices in East Kent, Canterbury, UK
[3]Department of Clinical and Experimental Medicine, University of Surrey, Guildford, UK
[4]Department of Mathematics, University of Surrey, Guildford, UK
[5]Primary Care and Population Health, Stanford School of Medicine, Stanford, California, USA
[6]National Association for Hospice at Home, Fareham, UK
[7]NHS England, Wakefield, UK
[8]Primary Care Unit, Department of Public Health and Primary Care, University of Cambridge, Cambridge, UK

**Correspondence to**
Professor Claire Butler;
c.butler-779@kent.ac.uk,
clairesinnott@aol.com

## ABSTRACT

**Introduction** Hospice at home (HAH) services aim to enable patients to be cared for and die in their place of choice, if that is at home, and to achieve a 'good death'. There is a considerable range of HAH services operating in England. The published evidence focuses on evaluations of individual services which vary considerably, and there is a lack of consistency in terms of the outcome measures reported. The evidence, therefore, does not provide generalisable information, so the question 'What are the features of hospice at home service models that work, for whom, and under what circumstances?' remains unanswered. The study aims to answer this question.

**Methods and analysis** This is a mixed-methods study in three phases informed by realist evaluation methodology. All HAH services in England will be invited to participate in a telephone survey to enable the development of a typology of services. In the second phase, case study sites representing the different service types will collect patient data and recruit carers, service managers and commissioners to gather quantitative and qualitative data about service provision and outcomes. A third phase will synthesise and refine the results through consensus workshops.

**Ethics and dissemination** The first survey phase has university ethics approval and the second phase, Integrated Research Application System (IRAS) and Health Research Authority (HRA) approval (IRAS ID:205986, REC:17/LO/0880); the third phase does not require ethics approval. Dissemination will be facilitated by project coapplicants with established connections to national policy-making forums, in addition to publications, conference presentations and reports targeted to service providers and commissioners.

## INTRODUCTION

Hospice at home (HAH) services have evolved in England since the development of the 'modern hospice movement' in the late 1960s. These services tend to share the following characteristics:
► Aim to enable patients to be cared for and die in their place of choice, if that is their own home.

### Strengths and limitations of this study

► The realist evaluation approach enables the complexity of different hospice at home models to be 'unpacked' within their context to understand what features work for whom and in what circumstances.
► Involving palliative patients at the end of life and their carers will be challenging in relation to approaching participants at this sensitive time and potential loss to follow-up of carers who are subsequently bereaved.
► In-depth case studies of up to six hospice at home models with different features will be part of the realist evaluation, therefore it is possible that not all feature combinations identified in the model typology from the survey of hospice at home services will be included.
► Patients without an informal carer involved on a daily basis will be excluded from the study.
► Patients and carers who are unable to communicate in English will be excluded from the study.

► Employ 'specialist' staff with high levels of palliative care experience.
► Ability to provide more staff time with the patient than pre-existing/other services.

A review of the literature identified service development projects and evaluations of services in England that have HAH characteristics.[1–31] Each study focused on an individual service and used various methods to investigate locally determined patient, carer and professional outcomes. Outcomes frequently focused on one or more of the following: place of death, fulfilment of wishes, carer satisfaction, carer bereavement, symptom management, experience of the service, hospital admission.

The literature mirrors the fact that different HAH services have grown up in an ad hoc fashion resulting in a considerable range of HAH services in terms of operation, staffing and function, making it difficult to identify

similar services in comparable settings. There has been little consensus as to what standards characterise such a service or what makes a service more or less effective. The National Association for Hospice at Home (NAHH) has recommended six core, national standards for HAH services developed through three national HAH stakeholder workshops held in 2011–2012.[32] The NAHH also worked with Hospice UK and conducted a survey across 76 HAH services in England which provided some useful data to start to describe the landscape of HAH services. This survey confirmed that more than one model of HAH service exists, and they are not homogeneous in their activities or outcomes.[33]

The best way to provide care within a patient's home and how this can be maintained for as long as possible was identified as one of the top 10 research priorities (in the UK) in a James Lind Alliance priority-setting partnership on palliative and end-of-life care published in January 2015.[34] It has been recognised that most people have a preference to die at home[35] and indeed the number of patients wishing to die at home is increasing.[36–38] A cost analysis from one study found that users of the HAH service had significantly lower utilisation of hospital services.[24] Given that almost half of annual deaths in England take place in National Health Service (NHS) hospitals,[39] it seems there is potential to increase the number of patients accessing community care and at the same time reduce NHS acute care costs. Demographic studies predict a future of increasing numbers of older people and increasing numbers of deaths.[39] A recent Health Ombudsman report highlighted how more needs to be done to support the health service in delivering quality care at the end of life.[40] HAH services offer an acceptable solution to meet these social and political drivers, yet their expansion has so far been haphazard. It is therefore important to understand how best to deliver effective HAH services at scale and in a cost-effective manner to achieve the outcomes desired.

No study comparing different types of HAH services has been identified. The variation in services and the settings in which they operate makes it difficult to conduct traditional comparative analyses and to achieve a meaningful synthesis of evidence which would help inform future service development and planning. This paper presents the protocol of a funded study running from 1 February 2017 to 31 January 2020 which aims to fill this evidence gap.

### Aims and objectives
The study's aim is to investigate the impact of the organisation and delivery of different models of HAH on patient and carer outcomes and experiences of end-of-life care from the perspective of service users, their family carers, service providers and commissioners. Our overarching research question is: What are the features of HAH service models that work, for whom and under what circumstances?

Objectives to address the primary research question are as follows:

1. Identify the range and variation of HAH models operating across England.
2. Categorise the models by type, setting and key features.
3. Select case studies of each model to enable an assessment of the impact of that model on patient and carer outcomes.
4. Investigate the resource implications and costs of patient care in each model.
5. Explore the experiences of patients, family carers, providers and commissioners of the different HAH models.
6. Identify the enablers and barriers to embedding HAH models as part of service delivery for end-of-life care.

HAH is a complex intervention and part of a whole system of health and social care delivery. The research design is informed by realist evaluation,[41 42] a theory-driven methodology increasingly used to evaluate complex interventions[43] including services for end-of-life care.[44] At the core of realist evaluation is the notion of 'generative mechanisms'; a generative mechanism is a causal link, the black box which leads from A to B and creates an 'effect'.[45] Realist evaluation theorises what the mechanisms are, the relationship between mechanisms, the context in which they are operating and the effects they produce through propositions, which take on a basic formula of: context+mechanism=outcome (CMO). The aim is to identify patterns to support an explanatory theory about what mechanisms are working (or not) in a given situation.[46] Data are sought to prove, refute and ultimately refine the conjectured CMO configurations.

## METHODS AND ANALYSIS
The method comprises three phases:

### Phase 1: telephone survey
A national telephone survey of all adult HAH services in England. There were 127 services identified from the Hospice UK Service Directory, cross-referenced with the NAHH database (received 28/10/2016) and contact will be made with service managers. The purpose of the survey is to produce a comprehensive map of the range and variation of HAH services and to group them into service model types, sharing similar characteristics. The survey will elicit information on service setting, configuration, operations and activity.

The interpretation of the survey findings will involve iterative consensus work with the project steering group and public and patient involvement advisory group to develop model types from the survey information. Categorical variables (eg, urban/rural, presence of hospice building(s), involvement of registered nurses (Yes/No), 24/7 care (Yes/No) etc) will be cross-tabulated with each other in order to identify underlying associations. Continuous variables (eg, area population, area (square miles),

number of individuals employed by the service) will be compared between different categories of each categorical variable, as well as being plotted against each other, in order to identify underlying associations. These results will assist in identifying natural groupings.

From this work, it is envisaged that at least four high-level types of HAH services will be distinguishable. This estimate is based on previous survey work which indicated there to be at least two types of model,[33] and an earlier literature review[47] in which we have found that there are services with and without registered nursing provision and services with and without the availability of rapid access 24/7.

## Phase 2: case studies

A sampling framework will be used to purposively select up to six case studies of HAH services from the high-level types identified in phase 1. Each type of model will be represented by at least one case study and the HAH service will be the unit of analysis. The approach will employ mixed methods to gain in-depth understanding of the impact of each type of model. This design also allows methodological flexibility to generate theoretical insights from the findings,[48] which is a key requirement for realist evaluative design.[42] The case studies will be used as test beds for candidate CMO configurations.[49] These candidate CMO configurations will be identified through our systematic mapping of the literature, our previous completed research and the NAHH core standards for HAH services.[32] We will also use normalisation process theory (NPT)[50] as a middle-range theory to understand how each HAH model becomes embedded within a whole system of care. NPT is focused on the 'work' that is involved in implementing a service, such as how staff make sense of the work they do and how they reflect on their practice.[51] NPT has previously been used successfully within an overall realist evaluative design to understand in detail the enabling contexts within CMO configurations.[52]

The impact of the HAH service on the following outcomes will be evaluated:

► The quality of death, using the quality of death and dying (QODD) tool,[53–55] completed with bereaved carer from 4 months post death, over the telephone; this is the primary outcome measure.

► Overall experience of care and support, using two questions selected from the National Survey of Bereaved People[56] and through in-depth interviews with a subset of family carers post bereavement.

► Whole system resource use gathered through interviews with carers every 2 weeks from recruitment to patient death using a customised version of the Ambulatory and Home Care Record (AHCR),[57] from which costs will be calculated.

► Service provider and commissioner views about enablers and barriers to delivering the HAH model, gathered through qualitative interviews. The service manager interviews will include items relevant to

understanding the economic costs of running each case study HAH service.

In addition, the Integrated Palliative Care Outcome Scale tool,[58] the Australia-modified Karnofsky performance scale and the phase of illness[59] will be completed by care professionals on entry to the HAH service to facilitate an understanding of the casemix of each service.

## Recruitment and informed consent

The study comes under the remit of the Mental Capacity Act 2005. Patients and their main lay carer within the case study sites will be invited to participate in the study when they are admitted to the HAH service. Service staff will introduce the study to the patient and their carer, provide written information and gain their consent if they agree to participate. Due to the nature of the patient population, who will be close to the end of life, it is anticipated that some of the potential participants will be unable to provide informed consent (eg, as a result of impaired cognition or impaired consciousness). For this reason, a variable consenting process, involving consultee assent, will be used. This method is increasingly used and accepted by NHS ethics committees as a process for gaining consent for patients who lack capacity.[60 61] If the patient is deemed not to have capacity, then a personal consultee (ie, 'someone who has a role in caring for the person who lacks capacity or is interested in that person's welfare but is not doing so for remuneration or acting in a professional capacity') will be approached for advice regarding the patient entering the study. The personal consultee could be a relation or a friend of the patient. The personal consultee will be given written information about the study and asked whether in their opinion the patient would have any objection to taking part. If the patient is deemed not to have capacity, and no personal consultee is available at the time or willing to take the responsibility, then a nominated consultee will be approached for advice regarding the patient entering the study; this is usually a health professional who knows the patient and is independent of the research team.

## Sample size
### Quantitative data collection sample size
The scores for the primary outcome measure, the QODD, range from 0 to 100. Hales *et al*[62] identified 30 and 70 as cut-offs for distinguishing terrible/poor, intermediate and good/almost perfect quality of death. On the basis of a difference of 10 points representing a meaningful change, and using an SD of 16.41,[63] at least 44 participants in each model would be required for comparisons between any pair. To allow for participant drop-out of 33%, the required sample size is 66 patients per model type. The drop-out rate is based on a prospective trial of an intervention which followed up with the carers of patients involved who were sent the 24-item intensive care unit QODD questionnaire 4–6 months post death. They received a 55.4% response rate and it is predicted that contact through bereavement services and the telephone

interview approach will achieve a better response than the postal survey approach used in that study.[64]

### Qualitative data collection sample sizes

Using a purposive sampling approach,[65] 5–10 managers, healthcare staff and commissioners in each case study site will be interviewed.

It is anticipated that up to 20 bereaved family carers per site will be interviewed until data saturation is reached to explore experiences of service use, particularly what aspects of service provision contribute to positive and negative experiences of care in order to understand what aspects of care are most valued by service users.

## Data analysis

### Quantitative data analysis

The characteristics of patients in the different service models will be summarised using relevant descriptive statistics (proportions, medians, ranges, means, SDs, 95% CIs, etc) before being compared on the basis of each patient's sociodemographic, clinical and carer features using the appropriate bivariate test (including one-way analysis of variance, $X^2$ and Kruskal-Wallis tests, depending on the nature of the variable). Exploratory regression modelling will be used in order to investigate the effect of each service model on the primary outcome (QODD), controlling for sociodemographic, clinical and carer features. Stepwise regression methods (backward elimination approach, commencing with a set of covariates which have been agreed on as important by the research team) will be used. The fitted parameters in the final models, along with their significance, will indicate if service type is associated with differences in QODD scores.

### Economic analysis

The economic analysis will be at two levels:

First, a descriptive analysis will be conducted of the resources and costs of running each case study HAH service. Information on staffing and activity rates will enable costs per patient receiving HAH to be calculated.

Second, a patient-level analysis will be undertaken. Whole system resource use in the end-of-life care will be captured prospectively from the point of recruitment to the study for each patient using a customised version of the AHCR, administered by telephone to the carer every 2 weeks until death. Retrospective data will also be collected from the carer shortly after recruitment to cover the period before the HAH service started. Service utilisation data will cover primary, community, hospital, hospice, social care, voluntary and informal care received. The AHCR has been developed and widely used for assessing resource use in home palliative care in Canada, including carer burden. In a recent systematic review of approaches to capturing the financial costs of family care giving within a palliative care context, the AHCR was identified as the only validated tool covering formal and informal services.[66] The AHCR has been recently piloted

in England by the research team and found to be both acceptable to participants and sensitive. Experience in the pilot indicated how the AHCR will be customised for use in the British NHS context.

Service use data, once captured, will be grouped into 4–6 time periods of approximately equal sample size, delimited by survival time from the start of service use data collection. The cut points will be determined by the distribution of the data. In the research team's previous study, 6% of patients referred to a HAH service had died within 2 days, 40% within 1 month, 62% within 2 months and the remaining 38% were referred over 2 months before death. Resource use will be converted to costs using national tariffs.[67] Informal care will be valued using replacement cost methods.

For each of the models of HAH service provision, an average cost per day of treatment will be estimated for the 4–6 time periods. Costs will be presented as means and median, given the typical skew in the distribution of costs. Comparison of costs between HAH models will be assessed for significance using the Mann-Whitney U test. Sensitivity analysis for costs will be handled deterministically, varying the amount of resource use between their upper and lower limits for each HAH model. Costs will be analysed in relation to outcomes from different models in a cost-consequences framework.

### Qualitative data analysis

Interviews will be transcribed and uploaded into NVivo to assist with data management and analysis. Analysis will be iterative with the aim of testing and refining programme theories and further developing provisional CMO configurations.[42] As described above, NPT will be used to understand why a model has or has not been embedded within a whole system of care, and burden of treatment will be used to understand the impact of the model on patients and carers. NPT offers a well-established framework for analysis in order to understand implementation processes through the perspectives of multiple stakeholders, including: service users, service providers and commissioners.[68] Constructs from the NPT framework will form the basis of a deductive coding structure. Analysis will also seek to identify any emergent themes not covered by NPT. Synthesis of an NPT informed coding framework alongside an inductive approach[69] allows for a focused and yet open qualitative approach that allows unexpected findings to emerge.[68]

### Phase 3: stakeholder consensus

The final phase comprises two national consensus workshops, with up to 60 participants attending each workshop. Participants will be identified through the NAHH and our project steering group. Stakeholders will include service providers, commissioners and service user representatives. The purpose of the workshops is to validate interpretation of the data and to refine our understanding of the specific features of HAH models that work, for whom and under what circumstances.

Emerging findings and relationships between CMOs will be presented to stakeholders.[42] The explicit aim of the workshops will be to refine CMO configurations and develop consensus on what type of HAH services are likely to work best and in what circumstances. The workshops will also contribute to translating findings into information that is relevant to managers and commissioners of HAH services. Consensus workshop methods will be used[70] to facilitate discussion. Detailed notes of the discussions will be written and used to verify or challenge CMO configurations. Participants will be sent a workshop report and have the opportunity to comment on the study conclusions.

## Synthesis

Data analysis from each phase of the study will be synthesised through a realist evaluative process comprising four stages[71]:

### Stage 1

Articulation of programme theories and propositions from literature review already undertaken, stakeholder insight (study steering group and service users) and phase 1 of the study (national telephone survey). Identification of candidate CMO configurations.

### Stage 2

Data collection from the model case studies in phase 2 of the study, to test and refine propositions.

### Stage 3

Map the outcomes of each model including costs; interrogate what contexts and mechanisms explain the pattern of outcomes.

### Stage 4

Through stakeholder consensus (phase 3 of the study), refine explanatory CMO configurations to evaluate what HAH model works best, for whom and in what circumstances.

## Patient and public involvement

A lay advisory group was set up through a local hospice to inform the development of the study design, including feedback on the project idea, research question and outcome measures, reviewing funding application drafts and the plain English summary. The group consisted of four members, including two bereaved carers and two members of the public. The bereaved carers had previously had direct experience of HAH service as carers for patients receiving the service. The members of the public (one being a hospice volunteer) had a keen interest in research and the work of the hospice.

Membership of the group to support the study development had been advertised to carers, patients and members of the public locally but no patients came forward. This was not unexpected as experience from a previous study showed that palliative patients found it difficult to participate continuously due to ill health. Involvement from the bereaved carers was key, as carers are the main participants in the study, and these group members went on to become part of the research team as lay coapplicants on the research grant application.

The lay advisory group members were involved in the design of the study through face-to-face meetings and one-to-one via email. They influenced the design of the study, and for phase two in particular, they were able to feedback on the appropriateness of the data collection tools and what procedures should be in place when approaching and involving patients, carers and bereaved carers in the research, at a vulnerable time on a sensitive topic. Examples of their input are: the inclusion of additional support from research staff to help carers with the completion of the service use information and the provision of information for carers to access further advice and support from the hospice should they need it following the completion of the QODD questionnaire and interview. During the course of the project the coapplicant members will be invited to provide feedback on the design of study materials, including information sheets, study leaflet and study outputs, such as lay summaries to disseminate the results to study participants. As patients who have capacity will be invited to participate, feedback on the patient version of the information sheet will be sought from current hospice patients through the hospice day-care service. The lay coapplicants will be invited to input on decision-making as stakeholders in the consensus of the CMO configurations for the realist evaluation. Appropriate training will be provided by the University of Kent and the lay coapplicants will receive payment to cover their time and expenses. All members of the research team work with the lay coapplicants and advisory group and in addition, the role of one member of the team is focused on coordinating and facilitating patient and public involvement to ensure it is threaded throughout the project.

## ETHICS AND DISSEMINATION

The study has been designed so that the burden to patients will be minimal, requiring consent to the collection of data already recorded for clinical purposes and consent to invite their carer to participate.

The main burden will fall on informal carers, to collect service utilisation data and then to respond to the telephone questionnaires administered postbereavement. Those who agree in addition to participate in a qualitative interview will be consented again and may find the interview process emotional and an opportunity to express their views about their experiences. The following steps have been taken to mitigate the burden for carers: the postbereavement data will be collected at 4 months; research staff will be trained and supported by the chief investigator, an experienced palliative care clinician and will be sensitive to the feelings of participants; interviews will be terminated if significant signs of distress develop and there is a distress protocol to provide support and follow-up.

*Phase 1 approvals:* NHS Health Research Authority approval reference 17/HRA/0299. SRC ethics panel of the University of Kent School of Social Policy, Sociology and Social Research, approval 13/12/16.

*Phase 2 approvals:* National Research Ethics Service, London—Queen Square Research Ethics Committee, IRAS 205986; REC:17/LO/0880.

*Phase 3:* Interpretation and dissemination of findings, ethics approval not required.

### Safety considerations

Research interviews may involve lone workers undertaking interviews at a carer's home. There are standard university policies and procedures for such situations which will be followed.

### Data deposition and curation

Throughout the study we will be fully compliant with the provisions of the Data Protection Act 1998, Human Rights Act 1998, NHS Code of Practice on Confidentiality and Common Law of Confidentiality. Participants' confidentiality will be ensured by using unique, untraceable identification code numbers to correspond to electronic data in the computer files.

All names will be anonymised and will be stored as a password-protected electronic file exclusively on University of Kent PCs/servers. Passwords will be restricted to: the chief investigator, the project manager and research personnel who directly engage with participants for the purposes of collecting data.

### Dissemination plan

Our primary output will be guidelines for services and commissioners to guide resource allocation and service development of HAH services. The guidelines will show what models/features of HAH services work best and at what cost and enable providers and commissioners to identify what the optimum HAH service model or key features of a HAH service would be for their population in their locality and organisational systems. The format of this guidance will be informed as part of the consensus workshops in phase 3 of the study. Additionally, the consensus events themselves will offer the opportunity for service providers to come together to share challenges and discuss good practice.

### Policy-maker, commissioner and professional engagement

We aim to reach commissioners, palliative care service providers and a wider professional audience through strong coapplicant links with existing forums (eg, the Commissioning Assembly, the NAHH, Hospice UK) and through publication in health services journals and conference presentations.

### Written publications

The full and complete account of the research will be published in the NIHR HS&DR Journal; this will allow the research to be freely and publicly available via the NIHR journals library website. We also aim to publish in peer-reviewed journals to reach broad audience coverage in community services as well as palliative care services.

A plain English summary for public and patient engagement and dissemination will be written and this will also be available to our research participants.

### Presentations

Oral presentations will be submitted to existing research forums such as the European Association of Palliative Care Congress; Clinical Research Network forums; Cicely Saunders Institute, King's College London; Hospice UK annual conference; NAHH conference.

### Social media

We will use twitter (#opelstudy) throughout the project via the Centre for Health Services Studies twitter account (@CHSS_Kent) to update on progress and debate, including discussions at the consensus event.

### Public

Dissemination of findings aimed at the public will be facilitated through links with organisations including the National Council for Palliative Care and Dying Matters.

**Contributors** CBu prepared this manuscript, led the research team and supervised the development of the protocol. CBr undertook the literature review and coordinated the writing of the protocol. HG devised the economic evaluation methodology. PW devised the statistics methodology. KG provided advice and context about hospice at home services in England to enable the development of the methodology. BW supported the development of the dissemination strategy. SB supported the development of the methodology from a primary care clinical perspective with his experience of previous research in this area. PW proposed the realist evaluation methodology and with LH devised the qualitative data methodology.

**Funding** The study has been funded by the Health Services and Delivery Research programme of the National Institute for Health Research (Reference 14/197/44).

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
