## [Reviewer comments · BMJ Open]

ARTICLE DETAILS

TITLE (PROVISIONAL)	Optimum hospice at home services for end of life care; protocol of a mixed methods study employing realist evaluation
AUTHORS	Butler, Claire; Brigden, Charlotte; Gage, Heather; Williams, Peter; Holdsworth, Laura; Greene, Kay; Wee, Bee; Barclay, Stephen; Wilson, Patricia

VERSION 1 – REVIEW

REVIEWER	Bridget Candy, Principal Research Fellow University College London, UK
REVIEW RETURNED	03-Jan-2018

GENERAL COMMENTS	The authors propose a very ambitious but needed piece of research. The multiple stages will be challenging and at each stage pragmatic decisions will need to be made, justified and documented. They may also find that some things they foresee in particular the four overall models may be difficult to achieve as an outcome. I provide a few comments on the text: ABSTRACT Can you reconsider these sentences 'The heterogeneous nature of the published evidence does not lend itself to systematic review to enable an distillation of the evidence to guide how best to deliver effective HAH services at scale and in a cost effective manner to achieve the outcomes desired. This study aims to fill the evidence gap.' Current it does not explain what is wrong with heterogeneity in a review and how your study can fill the gap. Can you provide a research question in your abstract. INTRODUCTION In the introduction you say 'Hospice at home services (HAH) have grown up in England since the development of then "modern hospice movement" in the late 1960s'. What do you mean by grown up? Page 5, LINE 15. Whose review, should you reference this? Although this sentence and the following are giving too much detail, does the reader need to know about funding and a previous review. Could these statements be simplified by just stating what studies you are aware of? Put NAHH in full first. Page 5, LINE 38 could say James Lind alliance was UK based
--

	survey. METHODS: Who is to be contacted at the HAH? T Interpretation of the survey – do you mean how to group it in a meaningful way into model types? How are the 6 case studies selected? Page 7, line 26 'The intervention to be measured is' – could you use another word than intervention, as it isn't really an intervention??
--	---

REVIEWER	Professor Barbara Jack Edge Hill University Ormskirk Lancs, UK
REVIEW RETURNED	09-Jan-2018

GENERAL COMMENTS	Thank you for asking me to review this excellent protocol. It is a very interesting study and very timely. There are a couple of very minor changes that I would suggest a) Abstract under methods and analysis line 15 states survey questionnaire. On page 5 line 41 states telephone survey- suggest change abstract b) Abstract HAH is abbreviated in line 1, under methods and analysis it is in full c) NAHH- pg4 line 30 need in full d) Sample size pg 7 line 25 drop out rate states sent the 24 item –questionnaire by telephone – suggest re phrase, is it post rather than telephone ? as that would fit with the line that follows on I wish the project team every success with the project and look forward to the findings
---

VERSION 1 – AUTHOR RESPONSE

Dear Dr Clark

Many thanks for your letter and I am pleased to revise the manuscript and resubmit for publication. I summarise below the response to the reviewer's comments and resulting amendments.

Editors comments:

The "strengths and limitations" section now reads:

- The realist evaluation approach enables the complexity of different hospice at home models to be 'unpacked' within their context to understand what features work for whom and in what circumstances.
- Involving palliative patients at the end of life and their carers will be challenging in relation to approaching participants at this sensitive time and potential loss to follow up of carers who are subsequently bereaved.
- Up to six in depth case studies of hospice at home models with different features will be part of the realist evaluation, therefore it is possible that not all feature-combinations identified in the model typology from the survey of hospice at home services will be included.
- Patients without an informal carer involved on a daily basis will be excluded from the study.

- Patients and carers who are unable to communicate in English will be excluded from the study.

The "Ethics considerations" and "Dissemination plan" have been combined as advised and ethical approval details included in this section which now reads:

ETHICS AND DISSEMINATION

The study has been designed so that the burden to patients will be minimal, requiring consent to the collection of data already recorded for clinical purposes and consent to invite their carer to participate. The main burden will fall on informal carers, to collect service utilisation data and then to respond to the telephone questionnaires administered post-bereavement. Those who agree in addition to participate in a qualitative interview will be consented again and may find the interview process emotional but also an opportunity to express their views about their experiences. The following steps have been taken to mitigate the burden for carers: the post bereavement data will be collected at 4 months; research staff will be trained and supported by the chief investigator, an experienced palliative care clinician, and will be sensitive to the feelings of participants; interviews will be terminated if significant signs of distress develop and there is a distress protocol to provide support and follow up

Phase 1 approvals:

NHS Health Research Authority approval reference 17/HRA/0299

SRC ethics panel of the University of Kent School of Social Policy, Sociology and Social Research, approval 13/12/16

Phase 2 approvals:

National Research Ethics Service, London - Queen Square Research Ethics Committee, IRAS 205986; REC:17/LO/0880

Phase 3:

No data collection or public involvement, ethics approval not required.

Safety considerations

Research interviews may involve lone workers undertaking interviews at a carer's home. There are standard University policies and procedures for such situations, which will be followed.

Data deposition and curation

Throughout the study we will be fully compliant with the provisions of the Data Protection Act 1998, Human Rights Act 1998, NHS Code of Practice on Confidentiality and Common Law of Confidentiality. Participants' confidentiality will be ensured by utilising unique untraceable identification code numbers to correspond to electronic data in the computer files.

All names will be anonymised and will be stored as a password protected electronic file exclusively on University of Kent PCs/servers. Passwords will be restricted to: the chief investigator, the project manager, and research personnel who directly engage participants for the purposes of collecting data.

Dissemination plan

Our primary output will be guidelines for services and commissioners to guide resource allocation and service development of HAH services. The guidelines will show what models/features of HAH services work best and at what cost and enable providers and commissioners to identify what the optimum HAH service model or key features of a HAH service would be for their population in their locality and organisational systems. The format of this guidance will be informed as part of the consensus workshops in phase 3 of the study. Additionally, the consensus events themselves will offer the opportunity for service providers to come together to share challenges and discuss good practice.

Policymaker, commissioner and professional engagement. We aim to reach commissioners, palliative care service providers and a wider professional audience through strong co-applicant links with existing forums (e.g. the Commissioning Assembly, the NAHH, Hospice UK) and through publication in health services journals and conference presentations.

Written publications

The full and complete account of the research will be published in the NIHR HS&DR Journal; this will allow the research to be freely and publicly available via the NIHR journals library website. We also aim to publish in peer reviewed journals to reach broad audience coverage in community services as well as Palliative care services.

A Plain English summary for public and patient engagement and dissemination will be written and this will also be available to our research participants.

Presentations

Oral presentations will be submitted to existing research forums such as the European Association of Palliative Care Congress; Clinical Research Network forums; Cicely Saunders Institute, King's College London; Hospice UK annual conference; National Association for Hospice at Home (NAHH) conference.

Social media

We will use twitter (#opelstudy) throughout the project via the Centre for Health Services Studies twitter account (@CHSS_Kent) to update on progress and debate, including discussions at the consensus event.

Public

Dissemination of findings aimed at the public will be facilitated through links with organisations including the National Council for Palliative Care and Dying Matters.

Reviewers' comments

Reviewer 1

The abstract has been rewritten to address these comments and the relevant section now reads as follows:

ABSTRACT

Introduction: Hospice at home (HAH) services aim to enable patients to be cared for and die in their place of choice, if that is at home, and to achieve a "good death". There is a considerable range of hospice at home services operating in England. The published evidence focuses on evaluations of individual services, which vary considerably, and there is a lack of consistency in terms of the outcome measures reported. The evidence therefore does not provide generalisable information, so the question "what are the features of hospice at home service models that work, for whom, and under what circumstances?" remains unanswered. The study aims to answer this question.

INTRODUCTION, the term "grown up" has been replaced by "evolved".

Page 5, line 15: as recommended, the report of the literature review has been simplified and clarified and now reads: A review of the literature identified service development projects and evaluations of services in England that have HAH characteristics.(1-31)

NAHH has been put in full, the National Association for Hospice at Home.

Page 5, line 38 now includes reference to the fact that the James Lind alliance survey identified research priorities in the UK.

METHODS: now specified as follows: There were 127 services identified from the 'Hospice UK' Service Directory, cross-referenced with the National Association for Hospice at Home database (received 28/10/2016) and contact will be made with service managers.

Interpretation of the survey - this has now been reworded as follows: The purpose of the survey is to produce a comprehensive map of the range and variation of HAH services and to group them into service model types, sharing similar characteristics.

How are the 6 case studies selected? This is described in the original submission as follows: "From this work it is envisaged that approximately 4 high-level types of HAH services will be distinguishable. This estimate is based on previous survey work which indicated there to be at least two types of model(33), and an earlier literature review(47) in which we have found that there are services with and without registered nursing provision and services with and without the availability of rapid access 24/7.

A sampling framework will be used to purposively select up to 6 case studies of HAH services from the high-level types identified in Phase 1. Each type of model will be represented by at least one case study and the HAH service will be the unit of analysis."

The details of the sampling framework will not be known until the phase 1 survey data has been collected and analysed.

page 7, line 26 - this has been amended as suggested and now reads: The impact of the HAH service on the following outcomes will be evaluated:

Reviewer 2

- a) Abstract has been amended to read "telephone survey" as suggested
- b) HAH abbreviation now used in Abstract, Methods and Analysis section as suggested.
- c) NAHH now in full as suggested.
- d) This was a postal questionnaire, now amended to: The drop-out rate is based on a prospective trial of an intervention which followed up with the carers of patients involved who were sent the 24 item ICU QODD questionnaire 4-6 months post death.